# Multi-Class Double-Transformation Network for SAR Image Registration

**Xiaozheng Deng** [1,*], **Shasha Mao** [2], **Jinyuan Yang** [2], **Shiming Lu** [2], **Shuiping Gou** [2], **Youming Zhou** [1] **and Licheng Jiao** [2]

[1] Chinese Flight Test Establishment, Xi'an 710089, China; zym3211@sohu.com
[2] Key Laboratory of Intelligent Perception and Image Understanding of Ministry of Education, School of Artificial Intelligence, Xidian University, Xi'an 710071, China; ssmao@xidian.edu.cn (S.M.); yangtuanzi@163.com (J.Y.); xdyaoshi@gmail.com (S.L.); shpgou@mail.xidian.edu.cn (S.G.); lchjiao@mail.xidian.edu.cn (L.J.)
*   Correspondence: dengxiaozheng11@163.com; Tel.: +86-1779-164-2506

**Abstract:** In SAR image registration, most existing methods consider the image registration as a two-classification problem to construct the pair training samples for training the deep model. However, it is difficult to obtain a mass of given matched-points directly from SAR images as the training samples. Based on this, we propose a multi-class double-transformation network for SAR image registration based on Swin-Transformer. Different from existing methods, the proposed method directly considers each key point as an independent category to construct the multi-classification model for SAR image registration. Then, based on the key points from the reference and sensed images, respectively, a double-transformation network with two branches is designed to search for matched-point pairs. In particular, to weaken the inherent diversity between two SAR images, key points from one image are transformed to the other image, and the transformed image is used as the basic image to capture sub-images corresponding to all key points as the training and testing samples. Moreover, a precise-matching module is designed to increase the reliability of the obtained matched-points by eliminating the inconsistent matched-point pairs given by two branches. Finally, a series of experiments illustrate that the proposed method can achieve higher registration performance compared to existing methods.

**Keywords:** synthetic aperture radar; SAR image registration; Swin-Transformer; double-transformation network





## 1. Introduction

Due to the special characteristics of synthetic aperture radar (SAR) [1], SAR image processing has been widely used in many fields [2,3]. In particular, some applications [4–8], such as SAR image change detection [4], SAR image fusion [5,6], open set recognition [9], etc., require the simultaneous processing of two or more SAR images. However, SAR images are generally obtained by different sensors under different conditions (such as time, viewpoint, noise). This leads to some differences among these images. Therefore, the registration of two or more SAR images becomes indispensable and significant. SAR image registration [10–12] is used to match two SAR images by exploring the geometric transformation model between them, where two images are called the reference image and the sensed image. At present, many registration methods [13–17] have been proposed to achieve the registration of two SAR images, and they can be simply divided into the traditional registration methods [13,18,19] and the deep-learning-based registration methods [12,20–23]. Due to the prominent performance of deep learning, the deep-learning-based method of SAR image registration has recently received more attention compared to traditional methods.

In general, a registration-model-based deep learning is designed from the perspective of the two-classification problem, where the pair of matched-points and the pair of non-matched-points are considered as the positive category and the negative category, respectively, [12,16,21,22,24]. Therefore, in order to achieve better registration performance, it is expected that a mass of matched-point and non-matched-point pairs could be given and then fed to the deep model. However, compared to non-matched-point pairs, it is difficult to obtain many reliable matched-point pairs directly from two SAR images, which also limits the performance of the deep model.

Interestingly, we find that each point in an SAR image is essentially different and independent from others. This essential characteristic leads us to a worthwhile thought: why did we not directly consider multiple key points as multiple classes to construct a multi-classification deep model for SAR image registration, instead of a two-classification deep model? Recently, Mao et al. [25] proposed an adaptive self-supervised SAR image registration method, which utilized each key point as an independent instance to train the self-supervised deep model and then compared the latent features of each key point with other points to search for the final matched-point pairs. Therefore, inspired by these, we aim to design the SAR image registration method from the perspective of multiple classification (discriminating multiple key points), where each key point is considered as an independent class, abandoning the idea of constructing a two-classification model (discriminating matched and non-matched pairs).

Noticeably, for the SAR image registration model, we know that its purpose is to find $K$ matched-points ($\rho^*$) between $m$ key points ($\rho_R$) on the reference image and $n$ key points ($\rho_S$) on the sensed image. Given $\rho_R = \{p_1^r, p_2^r, \ldots, p_m^r\}$ and $\rho_S = \{p_1^s, p_2^s, \ldots, p_n^s\}$, if $p_i^r \in \rho_R$ and $p_j^s \in \rho_S$ are a pair of matched-points, it means that they are from the same location in two images, and the image information corresponding to them should also be consistent. In brief, the mathematical description of three sets is given as

$$\rho^* = \rho_R \cap \rho_S, \textbf{ but } \rho_R \neq \rho_S, \tag{1}$$

where $\rho^* \neq \varnothing$ and $\rho^* = \{p_1^*, p_2^*, \ldots, p_K^*\}$, $K \leq \min\{m, n\}$. Therefore, *if each key point is considered as a class, the set of matched-points $P^*$ is equivalent to the set of overlaps between $\rho_R$ (with $m$ classes) and $\rho_S$ (with $n$ classes).* This means that there are some overlapping classes in all classes, when each key point is considered as an independent class to construct the multiple classification model.

As we know, in a learning model, the categories of training and testing sets are generally consistent, whereas in SAR image registration, only overlapping key points between reference and sensed images are consistent, since these points are matched-points, and the rest of key points should be different. In the other words, if all key points ($m + n$ points) are considered as classes, our real independent classes should be $m + n - K$ (not $m + n$), but $K$ is unknown and must be obtained by the model. Between $m$ classes and $n$ classes, there are only $K$ same classes, and meanwhile there are ($m + n - K$) different classes. It brings a difficult problem: determining how to construct a multi-classification model based on ($m + n$) given key points for SAR image registration.

Based on the above analyses, in this paper, we propose a double-transformation network for SAR image registration from the perspective of multi-classification, which mainly includes a coarse-matching-module-based double network and a precise-matching module. Considering that there are overlapping points between key points from reference and sensed images, the proposed method first constructs two multi-classification sub-networks, respectively, based on key points from two images, to seek coarse matched-points. In each sub-network, key points from one image are used as classes to train a multi-classification network; meanwhile, key points from another image are used as the testing set, where Swin-Transformer is used as the basic network. Then, with two multi-classification sub-networks, the prediction of $m$ key points can be obtained by the training model with $n$ key points, and the predictions of $n$ key points are obtained by the other training model with $m$ key points. Since only partial points are matched between key points

(classes) from two images, this means that the predictions of two models are inaccurate, but some predictions should be consistent. Therefore, secondly, a precise-matching module is designed to seek these consistent predictions from the results of two models as matched-points. Then, the registration transformation is obtained based on the obtained matched-points. In addition, to weaken the effect of inherent differences between two SAR images, a strategy is used to convert key points from the sensed image into the reference image by an initial transformation matrix, and the reference image is used as the base image to capture the sub-image of each key point. Finally, experimental results illustrate that the proposed method can achieve higher registration performance than the state-of-the-art methods.

Compared with most existing methods, the contributions of the proposed method are listed as follows:

- We utilize each key point directly as a class to design the multi-class model of SAR image registration, which avoids the difficulty of constructing the positive instances (matched-point pairs) in the traditional (two-classification) registration model.
- We design the double-transformation network with the coarse-to-precise structure, where key points from two images are, respectively, used to train two sub-networks that alternately predict key points from another image. It addresses the problem that the categories are inconsistent in training and testing sets.
- A precise-matching module is designed to modify the predictions of two sub-networks and obtain the consistent matched-points, where the nearest points of each key point are introduced to refine the predicted matched-points.

The rest of this paper is organized as follows. Section 2 first shows related works. Second, Section 3 introduces the detail of the proposed method. Then, Section 4 shows the corresponding experimental results and analysis to verify the performance of the proposed method. Finally, Section 6 shows the conclusion of this paper.

## 2. Related Works

### 2.1. The Attention Mechanism

As we know, people can receive a lot of information every day via various ways, whereas most people do not pay attention to all information, and they only selectively absorb a small amount of useful or interesting information. This ability is generally called attention. Inspired by the way human attention works, the researchers proposed the attention mechanism [26,27]. At present, the attention mechanism is widely used in natural language processing [28,29], image classification [30,31], object detection [32,33], etc.

Simply speaking, the attention mechanism selects the more critical information of the current target task by calculating the degree of correlation between the various samples. In general, the attention mechanism generates three vectors from the input vector, namely, the query vector $Q$ ($Q = [q_0, q_1, \ldots, q_m]$), key vectors $K$ ($K = [k_0, k_1, \ldots, k_m]$), and the value vector $V$ ($V = [v_0, v_1, \ldots, v_m]$). The attention mechanism is achieved by the following three steps.

First, the similarity of each key in the query vector $Q$ and the key vector $k_i$ is calculated by the formula [26,27]

$$s_i = sim(Q, k_i), \tag{2}$$

where $s_i$ expresses the similarity of the $i$th key, $i = 1, 2, 3, \ldots, m$. Here, the common similarity function can be used, such as Euclidean distance, the cosine similarity, etc. Then, the estimated similarity $s_i$ is normalized by the softmax function to obtain the weight $\alpha_i$ of the query vector and each key. Finally, the following formula is used to calculate the result of the weighted sum of all the elements in the weight $\alpha$ and the value vector $V$ as the attention vector $z_i$ [26,27], shown as follows:

$$z_i = \sum_{i=1}^{m} \alpha_i V_i. \tag{3}$$

In short, the attention mechanism works by calculating the weights between the query $Q$ and the corresponding key $K$, and then a weighted sum of all values in $V$ is followed as the final output.

### 2.2. The Transformer Model

Recently, Transformer [34] has been paid more attention due to its effectiveness, which mainly utilizes the self-attention mechanism to extract internal features and promote the performance of the network model. In general, there are encoder and decoder modules in the model of Transformer. In the encoder, the multi-head attention mechanism is the most important part, which allows the model to learn multiple feature representation spaces. In the decoder module, besides using the multi-head attention mechanism, the masked multi-attention mechanism is designed to hide the predicted information and meanwhile ensure that the results are predicted based on the known information each time.

For the tasks of computer vision and natural language processing, many studies [34–36] have illustrated the effective performance of Transformer. Vaswani et al. [34] introduced attention mechanism to solve machine translation and parsing tasks, which achieved good results. Devlin et al. [35] proposed a brand new linguistic representation model, which caused a stir in the NLP field, with the best results in many tasks in NLP. Thereafter, a wide variety of Transformer variants [37–39] were proposed. Dosovitskiy et al. [39] proposed Vision Transformer (ViT), which directly used the original Transformer structure and generated the sequence data of image blocks by cutting the image. The ViT model is directly used for image classification tasks, and can attain excellent results compared to state-of-the-art CNNs, whereas, when the ViT model is applied in vision tasks, there are still two main problems: one is that the feature size extracted by ViT is fixed, and the another is the that the computational complexity is high for the input image with large size.

The model of Swin-Transformer [40] was proposed by the Google team in 2021, which improves the ViT model [39] and further demonstrates the powerful performance of transformer in a series of visual tasks. Compared with ViT, Swin-Transformer achieves a layer-by-layer processing mechanism similar to the convolutional neural network, which reduces feature size gradually using patch merging to obtain multi-scale feature representations. In addition, it introduces locally aggregated information by performing the self-attention mechanism in the window [40], which is calculated by

$$\text{Attention}(Q, K, V) = \text{Softmax}(\frac{QK^T}{\sqrt{d}} + B)V. \tag{4}$$

The feature size of $Q$, $K$, and $V$ is $n \times d$, where $n$ is the number of token vectors in a window. $B$ is relative position bias, which is calculated by the relative position between the current patch index and the reference patch index.

Moreover, the Shifted-Window Multi-head Self-Attention (SW-MSA) module is also proposed, which is the improvement of the Window Multi-head Self Attention (W-MSA) module, and each Swin-Transformer block contains one W-MSA module and one SW-MSA module. The SW-MSA module can achieve the information interaction between windows while retaining the ability of reducing the computational complexity in the W-MSA module [40], shown as follows:

$$O(\text{W-MSA}) = (\frac{h}{k} * \frac{w}{k}) * (4k^2 C^2 + 2k^4 C^2) = 4hwC^2 + 2k^2 hwC^2, \tag{5}$$

where $h$ and $w$ represent the length and width of the input image, and $k$ is the length of Shifted-Window. In the Wwin-Transformer model, the width of the input image is $k * k$ rather than $h * w$; thus, there are $(h/M) * (w/M)$ windows to calculate. Due to the excellent performance of Transformer in image classification, this paper applies Swin-Transformer as the basic classification model to construct the multi-classification registration model for SAR images.

## 3. The Proposed Method

In this paper, we propose a multi-class double-transformation network of SAR image registration, where each key point is directly considered as an independence-class to construct the multi-classification model, named STDT-Net. The framework of STDT-Net is shown in Figure 1, which mainly includes two parts: the multi-class double-transformation networks based on multiple key points, and the precise-matching module to optimize the obtained pairs of matched-points. In the following, we will introduce each part in detail.

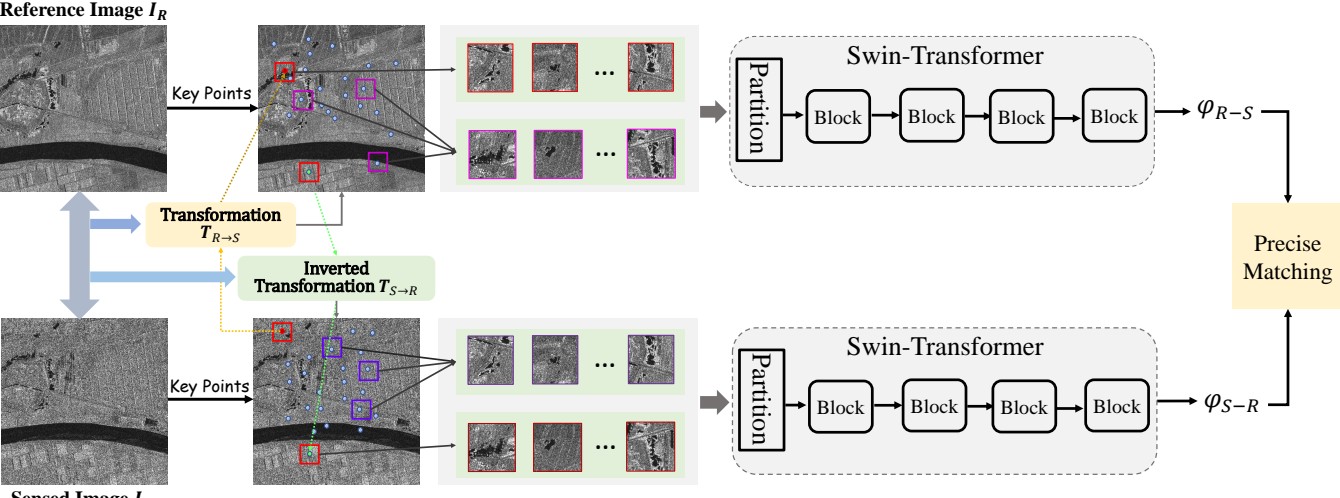

**Figure 1.** The framework of the proposed method.

### 3.1. The Multi-Class Double-Transformation Networks

In the proposed method, each key point in SAR images is considered as one category to construct the multi-classification model for SAR image registration. As we know, a pair of matched-points between two images are essentially located at the same coordinates. This means that two matched-points should belong to the same category and the non-matched-points should belong to different categories. Based on this, a double-transformation network is designed from the perspective of multiple classification to seek the coarse matched-points, which is composed of two branches, respectively, named the *R-S sub-network* and the *S-R sub-network*. In particular, the *R-S sub-network* uses the key points $\rho_R$ as the training set and the key points $\rho_S$ as the testing set, and the *S-R sub-network* uses the key points $\rho_S$ as the training set and the key points $\rho_R$ as the testing set.

#### 3.1.1. Constructing Samples-Based Key Points

For a classification model, the first crucial part is to construct the training and testing samples for the learning model. In order to construct the training and testing samples with multiple classification, two preparations need be implemented.

First, we use the SAR-SIFT algorithm [13] to detect key points from two SAR images, respectively. By SAR-SIFT algorithm, $m$ key points are obtained from the reference image, $\rho_R = \{p_1^r, p_2^r, \ldots, p_m^r\}$, and $n$ key points are obtained from the sensed image, $\rho_S = \{p_1^s, p_2^s, \ldots, p_n^s\}$. Then, the obtained key points are used as the centers to capture sub-images as training and testing samples.

In order to weaken the inherent diversities between two SAR images, a new strategy [25] is applied to transform key points from one image to the other image and then capture all sub-images (corresponding to all key points) from one same image. Here, a simple method is applied to obtain the initial transformation matrix as a rough registration. An example is given below to illustrate the details of constructing samples for two sub-networks. The R-S dataset corresponds to the *R-S sub-network*, and the S-R dataset corresponds to the *S-R sub-network*.

For the R-S date, the $m$ key points ($\rho_R$), detected from the reference image, are used as the centers to capture sub-images of size $s \times s$ from the reference image. Here, a sub-image corresponds to one sample. In addition, data enhancement is applied to enlarge training samples, mainly including the combination of rotation, brightness, cropping, etc., and each training sample is enhanced by $t$ times. Thus, based on $\rho_R$, a training set ($\mathcal{D}_{tr}^R$) is obtained, including $m * t$ training samples with $m$ classes, where each class corresponds to a key point in $\rho_R$ and of $t$ augmented samples (sub-images).

Then, for the testing samples, $n$ key points ($\rho_S$), detected from the sensed image, are used to construct their corresponding samples. In particular, to weaken the inherent diversities between two SAR images, we transform $\rho_S$ into the reference image by an initial transformation matrix (the initial transformation matrix $T_{R\text{-}S}$ can be given by one simple existing method; here, we use SAR-SIFT to obtain the initial transformation matrix) ($T_{R\text{-}S}$) and obtain their transformed key points ($\widetilde{\rho}_S^R$), where the transformation is given by

$$\widetilde{\rho}_S^R = T_{R\text{-}S}(\rho_S), \tag{6}$$

where $\widetilde{\rho}_S^R = \{\widetilde{p}_{s1}^r, \widetilde{p}_{s2}^r, \ldots, \widetilde{p}_{sn}^r\}$. Instead of $\rho_S$, the transformed points ($\widetilde{\rho}_S^R$) are used as the centers to capture sub-images from the reference image as the testing samples, where the size of each sub-image is $s \times s$. This give a testing set ($\mathcal{D}_{ts}^{\widetilde{S}}$) containing $n$ testing samples, where each sample corresponds to a key point in $\widetilde{\rho}_S^R$.

Similarly, for the S-R data, the $n$ key points ($\rho_S$) from the sensed image are used to obtain their corresponding training samples. With data enhancement, the training set ($\mathcal{D}_{tr}^S$) is obtained, including $n * t$ training samples with $n$ classes, where each class corresponds to a key point in $\rho_S$ and of $t$ samples (sub-images). Then, the $m$ key points ($\rho_R$) of the reference image are transformed to obtain their transformed points ($\widetilde{\rho}_R^S$), shown as

$$\widetilde{\rho}_R^S = T_{S\text{-}R}(\rho_R), \tag{7}$$

where $\widetilde{\rho}_R^S = \{\widetilde{p}_{r1}^s, \widetilde{p}_{r2}^s, \ldots, \widetilde{p}_{rm}^s\}$. Then, the transformed points ($\widetilde{\rho}_R^S$) are used as the centers to capture $m$ sub-images as testing samples. The testing set ($\mathcal{D}_{ts}^{\widetilde{R}}$) is obtained, including $m$ testing samples, where each sample corresponds to a key point in $\widetilde{\rho}_R^S$. Differently, all sub-images are captured from the sensed image for the S-R dataset, whereas all sub-images are captured from the reference image for the R-S dataset.

### 3.1.2. Multi-Class Double-Transformation Networks

In the proposed method, each key point is considered as a category to construct the multiple classification model. If $m$ key points ($\rho_R$) from the reference image are used as the training categories and $n$ key points ($\rho_S$) from the sensed image are used as the testing samples, the training model will be trained based on $m$ categories, and the $n$ testing samples will be predicted into $m$ categories by the training model. Obviously, this brings a category inconsistency problem, since only matched-points belong to the same category. Specifically, except for matched-points between $\rho_R$ and $\rho_S$, samples in the testing set should not be from any categories of the training set.

Hence, we design a double-transformation network that directly constructs two branches to train the multi-classification registration model, rather than a single branch network. In the proposed method, one branch is trained based on samples corresponding to $m$ key points from the reference image, and the other is trained based on samples corresponding to $n$ key points from the sensed image. Based on this, two datasets are constructed for two branches: the R-S data (for the R-S data, its training set $\mathcal{D}_{tr}^R$ contains $m * t$ samples with $m$ classes, and its testing set $\mathcal{D}_{ts}^{\widetilde{S}}$ contains $n$ samples), $\{\mathcal{D}_{tr}^R, \mathcal{D}_{ts}^{\widetilde{S}}\}$, and the S-R data (for the S-R data, its training set $\mathcal{D}_{tr}^S$ contains $n * t$ samples with $n$ classes, and its testing set $\mathcal{D}_{ts}^{\widetilde{R}}$ contains $m$ samples), $\{\mathcal{D}_{tr}^S, \mathcal{D}_{ts}^{\widetilde{R}}\}$, and the details are given in Section 3.1.1. Then, two multiple classification branches (sub-networks) are, respectively, constructed to find the matched-point pairs between reference and sensed images, defined as the R-S

branch ($Net_R$) and the S-R branch ($Net_S$), respectively. Here, the Swin-Transformer [40] is used as the basic learning network, as it has achieved the remarkable performance in many applications.

In the R-S branch, since $m$ key points are used as the categories of the multi-classification model, $n$ testing samples are predicted into $m$ categories, and a testing sample $\mathbf{x}_j^s$ is given a predicted class by $Net_R$, shown as

$$Pred_R(\mathbf{x}_j^s) = i, \ i = \{1, \ldots, m\}, \tag{8}$$

where $Pred_R(.)$ expresses the prediction result, and $\mathbf{x}_j^s \in \mathcal{D}_{ts}^{\widetilde{S}}$, $j = \{1, \ldots, n\}$. If a testing sample $\mathbf{x}_j^s$ is predicted as the $i$th category, it means that its corresponding key point ($p_{Sj}$) is possibly matched to the $i$th key point ($p_{Ri}$). Therefore, according to the prediction results, we can obtain a candidate set $\phi_i^r$ of matched-points for the $i$th key point ($p_i^r$) in $\rho_R$, shown as

$$\phi_i^r = \{(p_i^r, \widetilde{p}_{i1}^s), (p_i^r, \widetilde{p}_{i2}^s), \ldots, (p_i^r, \widetilde{p}_{iL_i}^s)\}, \tag{9}$$

where $L_i$ is the number of all testing samples that are predicted as the $j$th category. $\widetilde{p}_{il}^r$ corresponds to the $l$th testing sample ($\mathbf{x}_l^s$) which is predicted as the $j$th category by $Net_R$, shown as $Pred_R(\mathbf{x}_l^s) = i$, where $l = \{1, \ldots, L_i\}$. Therefore, we obtain all candidate sets $\{\phi_1^r, \phi_2^r, \ldots, \phi_m^r\}$ of matched-points for all key points $\rho_R$. Note that $\phi_j^s$ is an empty set if any testing samples are not predicted as the $j$th category.

Meanwhile, in the S-R branch, $n$ key points are used as the categories of the multi-classification model, and $m$ testing samples are predicted into $n$ categories. A testing sample $\mathbf{x}_i^r$ is given a predicted class by $Net_S$, shown as

$$Pred_S(\mathbf{x}_i^R) = j, \ j = \{1, \ldots, n\}. \tag{10}$$

Similarly, if the sample $\mathbf{x}_i^r$ is predicted as the $j$th category, it means that its corresponding key point ($p_{Ri}$) is possibly matched to the $j$th key point ($p_{Rj}$). Therefore, according to the prediction results, we can also obtain $n$ candidate sets of matched-points for $n$ key points in $\rho_S$, shown as $\{\phi_1^s, \phi_2^s, \ldots, \phi_n^s\}$, where

$$\phi_j^s = \{(p_j^s, \widetilde{p}_{j1}^r), (p_j^s, \widetilde{p}_{j1}^r), \ldots, (p_j^s, \widetilde{p}_{jL_j}^r)\}, \tag{11}$$

where $\widetilde{p}_{jl}^s$ corresponds to the $l$th testing sample ($\mathbf{x}_l^r$) that is predicted as the $j$th category by $Net_S$, shown as $Pred_S(\mathbf{x}_l^r) = j$, where $l = \{1, \ldots, L_j\}$.

As we know, the matched-points should be one-to-one, but the size $L_i$ (or $L_j$) of an candidate set is not equal to 1, since several testing samples are predicted to be in the same category. Based on this, we measure the similarities between the sub-image (corresponding to a key point $p_i^r$) and all sub-images (corresponding to its $L_i$ candidate matched-points), and then the most similar matched-points are selected as the final coarse matched-points obtained by two branches, defined as $\phi_{\text{R-S}}$ and $\phi_{\text{S-R}}$, respectively, shown in Figure 1.

### 3.2. The Precise-Matching Module

After obtaining the coarse matched-point pairs, a precise-matching module is designed to search for the more accurate and consistent matched-point pairs between two images, which utilizes the near-points of each key point to assist in estimating the matched-point pairs between the reference and sensed images. The details of the precise-matching module are introduced as follows.

According to [16], it is known that sub-images corresponding to different key points may be similar. Therefore, we first re-find eight near-points around each key point in the testing set. These near-points are located at the top left, top, right, left, left, right, bottom left, bottom and right, respectively, around the key point with $k$ pixels, and eight near-points are used as centers to capture their corresponding sub-images of size $s \times s$. Then, eight

sub-images are used as new testing samples and predicted by the training model to obtain the matched-point pairs of eight near-points. A simple example is shown in Figure 2.

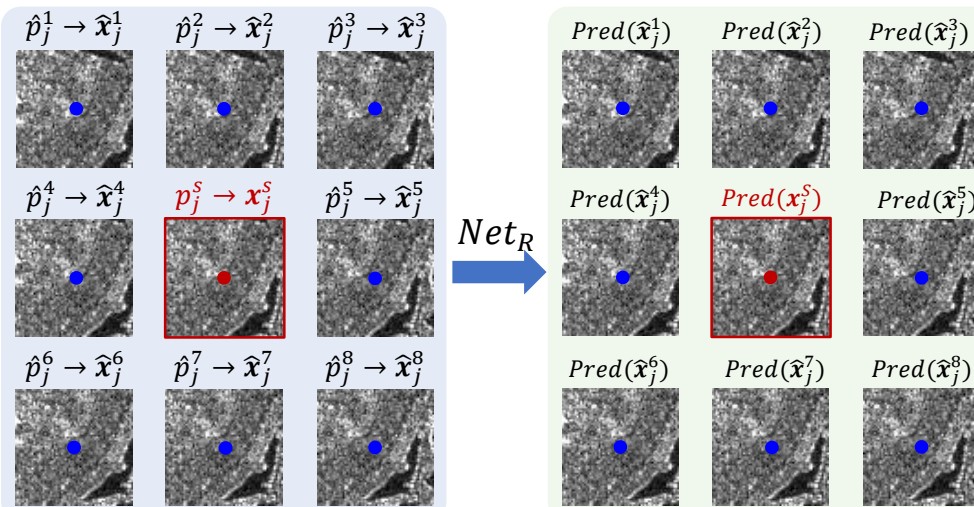

**Figure 2.** A visual example of eight near-points around a key point from the sensed image with $k$ pixels, where $k = 5$ and the predictions are obtained by the R-S branch ($Net_R$).

Second, for the pairs of coarse matched-points, we compare the matched-point pair of each key point in the testing set with the matched-point pairs of its eight near-points. If the predicted category of a key point is more than 50% consistent with the predicted categories of its eight near-points, we consider the prediction of that key point to be absolutely accurate. The consistency is calculated by

$$Cons(p_j^s) = \sum_{t=1}^{8} \mathbb{I}(Pred_R(\hat{x}_j^t) = Pred_R(x_j^S)), \tag{12}$$

where $x_j^S$ expresses the sub-image corresponding to the key point $p_j^s$, and $\hat{x}_j^t$ is the sub-image corresponding to the $t$th near-point of $p_j^s$. In contrast, if the consistence between a key point and its near-points is less than 50%, we consider that the prediction of this point may be slightly biased, and then delete its matched-point pair from $r$ pairs of matched-points. By this processing, the obtained matched-point pairs are updated by eliminating some incorrect matched-point pairs. Specifically, $\phi_{R\text{-}S}$ is updated as $\phi'_{R\text{-}S}$, and $\phi_{S\text{-}R}$ is updated as $\phi'_{S\text{-}R}$.

Then, based on the matched-point pairs $\phi'_{R\text{-}S}$ and $\phi'_{S\text{-}R}$, we merge the results obtained by two sub-networks to search for the overlapping pairs $\{(p_i^r, p_j^s)\}$. The overlapping pair $(p_i^r, p_j^s)$ are actually the consistent matched-points between two images. Therefore, the condition for two points to match is that they satisfy

$$Pred_R(p_j^s) = i \ \text{ and } \ Pred_S(p_i^r) = j, \tag{13}$$

where $i$ is the class corresponding to $p_i^r$ in the R-S branch, and $j$ is the class corresponding to $p_j^s$ in the S-R branch. If $p_j^s$ is predicted to be the $i$th category (corresponding to $p_i^r$) by $Net_R$, and meanwhile $p_i^r$ is predicted to be the $j$th category (corresponding to $p_j^s$) by $Net_S$, it means that two points, $p_j^s$ and $p_i^r$, are matched.

Moreover, since we use an initial registration to transform key points from one image to another in the process of sample construction, this may bring some bias in capturing sub-images based on the transformed point. Therefore, after obtaining the overlapping pairs of matched-points, we compare each key point with its near-points to select the most similar matched-point from nine points as the final overlapping point. Finally, the least

squares method [25,41] is used to calculate the transformation matrix based on the final overlapping matched-points.

## 4. Experiments and Analyses

In order to validate the registration performance of the proposed method, we implement experiments and analyses from four items: (1) comparing the registration performance of the proposed method with the state-of-the-art methods; (2) the visualization on chessboard diagrams of SAR image registration; (3) the analysis on the precise-matching module; (4) the analysis on the double-transformation network. In the experiments, four datasets are used to validate the registration performance of the proposed method, including Wuhan, Australia-Yama, YellowR1, and YellowR2 datasets, and more detailed descriptions are available in [16,25]. Four datasets are shown in Figures 3–6, respectively.

The Reference Image          The Sensed Image

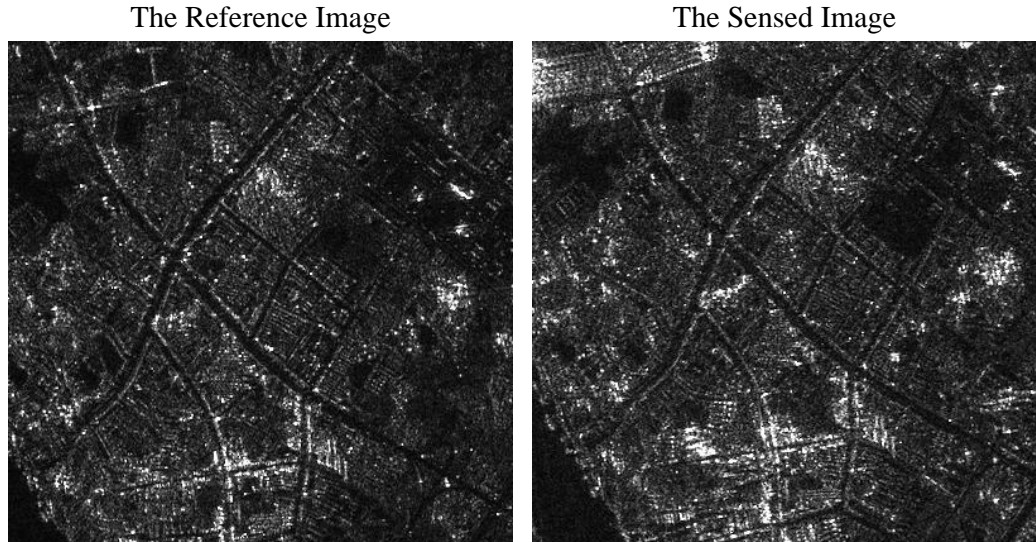

**Figure 3.** Reference and sensed images of Wuhan data. The image size is $400 \times 400$ and the resolution is 10 m.

The Reference Image          The Sensed Image

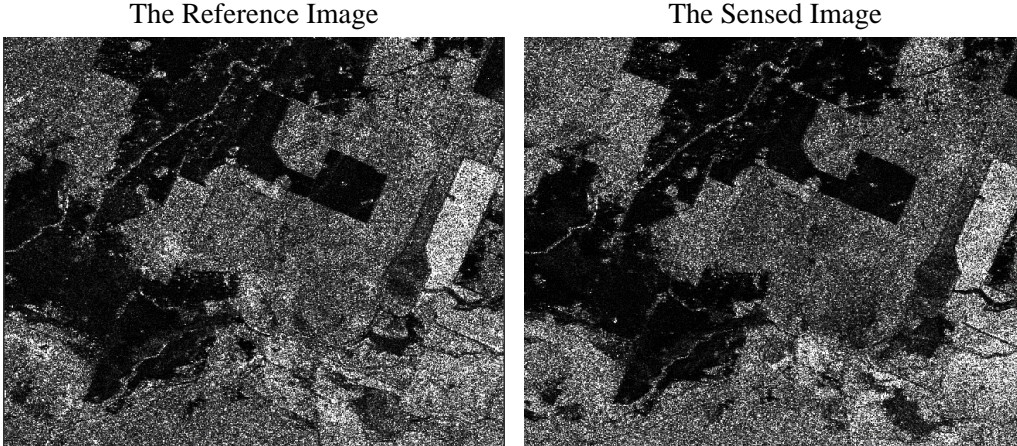

**Figure 4.** Reference and sensed images of Australia-Yama data. The image size is $650 \times 350$ pixels.

All experiments are implemented in the environment with NVIDIA GeForce RTX 2080Ti and Windows 10 with 64 GB memory and kernel Intel (R) Xeon (R) CPU E5-2605 v3 @2.30 GHz, and the frame is Pytorch. In the process of data enhancement, three transformation are used, including scale, rotation, and contrast. Note that the parameter settings are different for the training set and for the validation set. For the training samples, the parameter range of scale-transformation is $[0.5, 1.5]$, and the parameters of rotation-

transformation are selected between 1 to 20 degrees. Contrast-transformation is set at $p = 3$. For the validation sample, the parameter range of scale-transformation is $[0.5, 1.5]$, while the parameter of rotation-transformation is selected between 1 and 10 degrees. Meanwhile the contrast transformation is set at $p = 2$. For the parameters of Swin-Transformer, the batch size is set as 128, the feature-dim size is set as 128, and the temperature size is set as 0.5. Referred to [40,42], the layer number of each block is set as 2, 2, 18, and 2. An AdamW optimizer is used to train the network for 300 epochs with a cosine decay learning rate scheduler, the initial learning rate is set as 0.001, and the weight decay is set as 0.05.

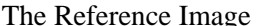

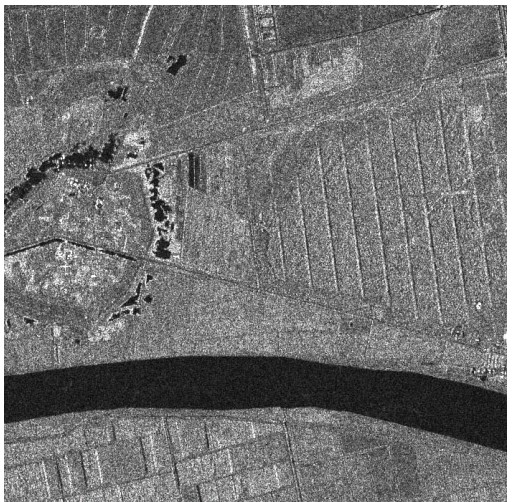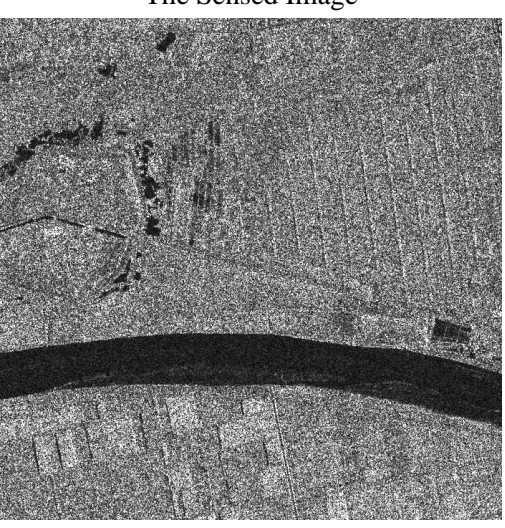

**Figure 5.** Reference and sensed images of YellowR1 data. The image size is $700 \times 700$ pixels and the resolution is 8 m.

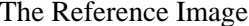

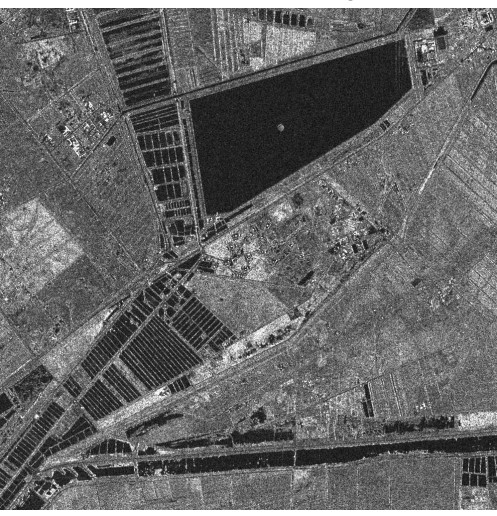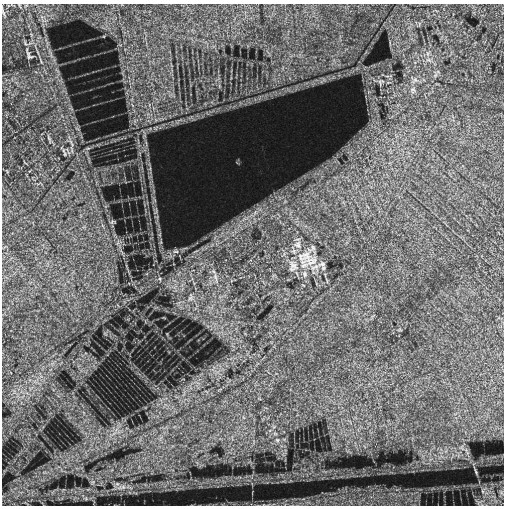

**Figure 6.** Reference and sensed images of YellowR2 data. The image size is $1000 \times 1000$ pixels and the resolution is 8 m.

In addition, eight quantified evaluation indicators [16] are used to validate the registration performance, including $RMS_{all}$, $RMS_{LOO}$, $N_{red}$, $P_{quad}$, $BPP(r)$, $S_{kew}$, $S_{cat}$, and $\phi$. Eight indicators are shown in detail as follows:

1.  $RMS_{all}$ expresses the root mean square error of the registration result. Note that $RMS_{all} \leq 1$ means that the performance reaches sub-pixel accuracy.
2.  $N_{red}$ is the number of matched-points pairs. Its value is higher, which may be beneficial for obtaining a transformation matrix with a better performance of image registration.

3. $RMS_{LOO}$ expresses the error obtained based on the Leave-One-Out strategy and the root mean square error. For each point in $N_{red}$, $RMS_{LOO}$ is the average of all errors ($RMS_{all}$ of $N_{red} - 1$ points).

4. $P_{quad}$ is used to detect whether the retained feature points are evenly distributed in the quadrant, and its value should be less than 95%.

5. $BPP(r)$ expresses the bad point proportion in obtained matched-points pairs, where a point with a residual value above a certain threshold ($r$) is called the bad point.

6. $S_{kew}$ denotes the absolute value of the calculated correlation coefficient. Note that the Spearman correlation coefficient is used when $N_{red} < 20$; otherwise, the Pearson correlation coefficient is applied.

7. $S_{cat}$ is a statistical evaluation of the entire image feature point distribution [43], which should be less than 95%.

8. $\phi$ is the linear combination of the above seven indicators, calculated by

$$\phi = \frac{1}{12}[2 \times (\frac{1}{N_{red}} + RMS_{LOO} + BPP(1.0) + S_{cat}) \qquad (14)$$
$$+ RMS_{all} + 1.5 \times (P_{quad} + S_{knew})].$$

When $N_{red} \geq 20$, $P_{quad}$ is not used, and the above formula is simplified as

$$\phi = \frac{1}{10.5}[2 \times (\frac{1}{N_{red}} + RMS_{LOO} + BPP(1.0) + S_{cat}) \qquad (15)$$
$$+ RMS_{all} + 1.5 \times S_{knew}],$$

and its value should be less than 0.605.

### 4.1. Comparison and Analysis of the Experimental Results

In this part, to validate the registration performance, we compare the proposed method with eight existing methods: SIFT [44], SAR-SIFT [13], VGG16-LS [45], ResNet50-LS [46], ViT-LS [39], DNN+RANSAC [12], MSDF-Net [16], and AdaSSIR [25]. In the nine compared methods, SIFT and SAR-SIFT are two traditional registration methods. **SIFT** is mainly matched by using the Euclidean distance ratio between the nearest and second-nearest neighbors of the corresponding features. **SAR-SIFT** is an improvement of the SIFT method, and it is more consistent with the SAR image characteristics. **VGG16-LS, ResNet50-LS and ViT-LS** are deep-learning-based classification methods. **DNN + RANSAC** [12] constructs the training sample set by using self-learning methods, and then it uses DNN networks to obtain matched image pairs. **MSDF-Net** [16] uses deep forest to construct multiple matching models based on multi-scale fusion to obtain the matched-points pairs, and then it uses RANSAC to calculate the transformation matrix. **AdaSSIR** [25] proposes an adaptive self-supervised SAR image registration method, where the registration of SAR images is considered as a self-supervised learning problem and each key point is regarded as a category-independent instance to construct the contrastive model for searching out the accurate matched points.

Tables 1–4 show the registration results on the four datasets, respectively. From four tables, it can be seen that the proposed method (STDT-Net) obtains better registration performances compared with other methods on four datasets. Obviously, the $RMS_{all}$ and $RMS_{LOO}$ of the proposed method are highest in six methods for all datasets, meanwhile obtaining the lowest bad point ratio ($BPP(r)$), whereas the final retained key points are not well distributed in the quadrant ($P_{quad}$). In short, the results for four datasets demonstrate that the proposed method can improve the performance of SAR image registration by double-transformation network based on Swin-Transformer.

**Table 1.** Comparison of registration performance obtained by six methods on Wuhan data.

| Methods | $N_{red}$ | $RMS_{all}$ | $RMS_{Loo}$ | $P_{quad}$ | $BPP(r)$ | $S_{kew}$ | $S_{cat}$ | $\phi$ |
|---|---|---|---|---|---|---|---|---|
| SIFT | 17 | 1.2076 | 1.2139 | — | 0.6471 | 0.1367 | 0.9991 | 0.7048 |
| SAR-SIFT | 66 | 1.2455 | 1.2491 | 0.6300 | 0.6212 | 0.1251 | 0.9961 | 0.6784 |
| VGG16-LS | 58 | 0.5611 | 0.5694 | 0.6665 | 0.2556 | **0.0389** | 1.0000 | 0.4420 |
| ResNet50-LS | 68 | 0.4818 | 0.4966 | 0.7162 | 0.2818 | 0.1943 | **0.9766** | 0.4489 |
| ViT-LS | 64 | 0.5218 | 0.5304 | **0.6101** | 0.2330 | 0.1072 | 1.0000 | 0.4296 |
| DNN + RANSAC | 8 | 0.6471 | 0.6766 | – | **0.1818** | 0.0943 | **0.9766** | 0.4484 |
| MSDF-Net | 39 | 0.4345 | 0.4893 | **0.6101** | 0.3124 | 0.1072 | 1.0000 | 0.4304 |
| AdaSSIR | 47 | **0.4217** | **0.4459** | 0.6254 | 0.3377 | 0.1165 | 1.0000 | 0.4287 |
| STDT-Net (Ours) | **78** | 0.4490 | 0.4520 | 0.6254 | 0.2277 | 0.1165 | 1.0000 | **0.4122** |
| Rank/All | 1/10 | 3/10 | 2/10 | 2/7 | 2/10 | 4/10 | 4/4 | 1/10 |

**Table 2.** Comparison of registration performance obtained by six methods on YAMBA data.

| Methods | $N_{red}$ | $RMS_{all}$ | $RMS_{Loo}$ | $P_{quad}$ | $BPP(r)$ | $S_{kew}$ | $S_{cat}$ | $\phi$ |
|---|---|---|---|---|---|---|---|---|
| SIFT | 69 | 1.1768 | 1.1806 | 0.9013 | 0.6812 | **0.0975** | 0.9922 | 0.7010 |
| SAR-SIFT | **151** | 1.2487 | 1.2948 | 0.6016 | 0.6755 | 0.1274 | 0.9980 | 0.6910 |
| VGG16-LS | 112 | 0.5604 | 0.5685 | 0.6150 | 0.3621 | 0.1271 | 1.0000 | 0.4626 |
| ResNet50-LS | 120 | 0.4903 | 0.5064 | **0.5873** | 0.2515 | 0.1027 | 1.0000 | 0.4215 |
| ViT-LS | 109 | 0.5276 | 0.5371 | 0.7162 | 0.2529 | 0.1105 | 1.0000 | 0.4472 |
| DNN+RANSAC | 8 | 0.7293 | 0.7582 | – | 0.5000 | 0.1227 | **0.9766** | 0.5365 |
| MSDF-Net | 12 | 0.4645 | 0.4835 | – | 0.4000 | 0.1175 | 0.9999 | 0.4356 |
| AdaSSIR | 71 | 0.4637 | **0.4707** | 0.6013 | 0.4545 | 0.1072 | 1.0000 | 0.4504 |
| STDT-Net (Ours) | 115 | **0.4604** | 0.4732 | 0.6740 | **0.2173** | 0.1175 | 1.0000 | **0.4205** |
| Rank/All | 3/9 | 1/9 | 2/9 | 5/7 | 2/9 | 4/9 | 4/4 | 1/9 |

**Table 3.** Comparison of registration performance obtained by six methods on YellowR1 data.

| Methods | $N_{red}$ | $RMS_{all}$ | $RMS_{Loo}$ | $P_{quad}$ | $BPP(r)$ | $S_{kew}$ | $S_{cat}$ | $\phi$ |
|---|---|---|---|---|---|---|---|---|
| SIFT | 11 | 0.9105 | 0.9436 | — | 0.5455 | 0.1055 | **0.9873** | 0.5908 |
| SAR-SIFT | **31** | 1.1424 | 1.2948 | 0.5910 | 0.7419 | 0.0962 | 1.0000 | 0.6636 |
| VGG16-LS | 19 | 0.6089 | 0.6114 | — | 0.4211 | 0.1061 | 1.0000 | 0.4703 |
| ResNet50-LS | 25 | 0.5725 | 0.5889 | 0.5814 | 0.6058 | 0.1387 | 1.0000 | 0.5102 |
| ViT-LS | 20 | 0.5986 | 0.5571 | 0.5821 | 0.5875 | 0.1266 | 1.0000 | 0.5118 |
| DNN+RANSAC | 10 | 0.8024 | 0.8518 | – | 0.6000 | 0.1381 | 0.9996 | 0.5821 |
| MSDF-Net | 11 | 0.5923 | 0.6114 | – | 0.4351 | **0.0834** | 0.9990 | 0.4753 |
| AdaSSIR | 20 | 0.5534 | 0.5720 | **0.5395** | 0.4444 | 0.1086 | 1.0000 | 0.4715 |
| STDT-Net (Ours) | 24 | **0.5487** | **0.5531** | 0.5486 | **0.4038** | 0.1088 | 1.0000 | **0.4610** |
| Rank/All | 3/9 | **1/9** | **1/9** | 2/7 | 1/9 | 6/9 | 4/4 | 1/9 |

**Table 4.** Comparison of registration performance obtained by six methods on YellowR2 data.

| Methods | $N_{red}$ | $RMS_{all}$ | $RMS_{Loo}$ | $P_{quad}$ | $BPP(r)$ | $S_{kew}$ | $S_{cat}$ | $\phi$ |
|---|---|---|---|---|---|---|---|---|
| SIFT | 88 | 1.1696 | 1.1711 | 0.6399 | 0.7841 | 0.1138 | **0.9375** | 0.6757 |
| SAR-SIFT | **301** | 1.1903 | 1.1973 | 0.8961 | 0.8671 | 0.1318 | 1.0000 | 0.7390 |
| VGG16-LS | 54 | 0.5406 | 0.5504 | 0.6804 | 0.3187 | 0.1277 | 1.0000 | 0.4607 |
| ResNet50-LS | 70 | 0.5036 | 0.5106 | 0.7162 | 0.2778 | 0.1208 | 0.9999 | 0.4470 |
| ViT-LS | 67 | 0.5015 | 0.5095 | **0.6000** | 0.2925 | 0.1281 | 1.0000 | 0.4356 |
| DNN+RANSAC | 10 | 0.5784 | 0.5906 | – | **0.0000** | 0.1308 | 0.9999 | **0.3946** |
| MSDF-Net | 52 | 0.5051 | 0.5220 | 0.6112 | 0.7692 | 0.1434 | 1.0000 | 0.5215 |
| AdaSSIR | 68 | 0.4858 | 0.4994 | 0.6013 | 0.5714 | 0.1149 | 1.0000 | 0.4776 |
| STDT-Net (Ours) | 79 | **0.4808** | **0.4954** | 0.6740 | 0.2692 | **0.1134** | 1.0000 | 0.4347 |
| Rank/All | 3/9 | 1/9 | 1/9 | 5/7 | 2/9 | 1/9 | 4/4 | 2/9 |

Compared with three compared classification networks (VGG16-LS, ResNet50-LS, ViT-LS, DNN+RANSAC, MSDF-Net, and AdaSSIR), the proposed method (STDT-Net) obtains a relatively large number of matching pairs ($N_{red}$) and the minimum error rate in obtaining the correct matching pair index, and also achieves better registration accuracy ($RMS_{all}$ and $RMS_{Loo}$). In general, the registration accuracy is better if a method has a large $N_{red}$ and a smaller $RMS_{all}$ value. It is seen that, compared with other classification networks, the obtained feature space is best when the Swin-Transformer network is used as the basic classification model. It means that the coordinate error between the obtained matched-points pairs is smaller, which results in the decreased proportion of bad points and the rise of $N_{red}$ indicators, meanwhile reducing the registration error. In short, the proposed method obtains better registration performance than other compared methods.

### 4.2. The Visual Results of SAR Image Registration

In this part, we draw the chessboard map (CB-map) of two matched SAR images to visually show the registration results on four datasets. Figures 7–10 show the CB-maps for four datasets, respectively. In each figure, the chessboard mainly focuses on the overlapping region of two image registrations, and the size of each checker is set based on the size of each datum. Except for the overlapping region, the other regions are filled by their corresponding images (reference or sensed image). For example, in Figure 9, the rightmost area of the CB-map is filled by the sensed image; meanwhile, the leftmost area of the CB-map is filled by the reference image. In order to enhance the visual effect, the contrast ratio between reference and sensed images is increased by darkening one image or brightening the other image.

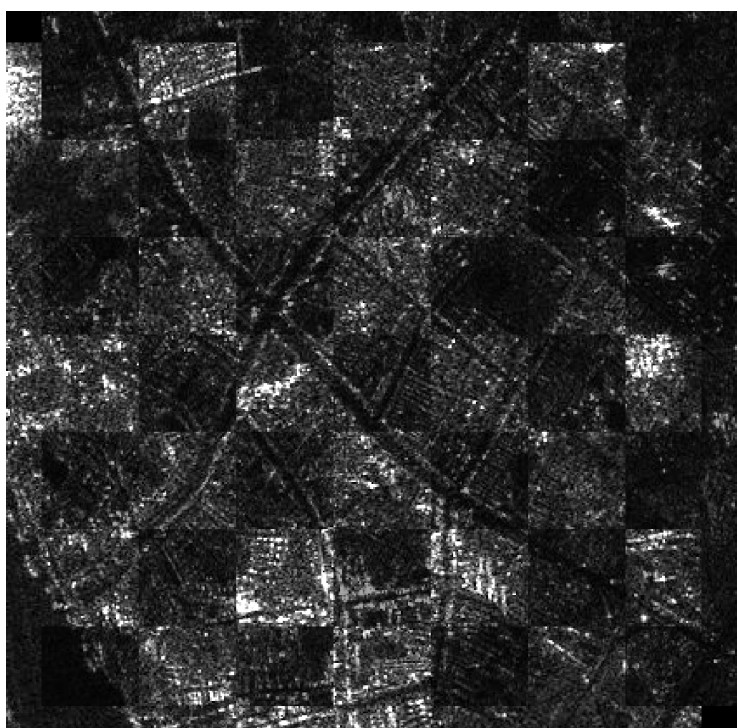

**Figure 7.** Registration CB-map for Wuhan data with 400 × 400.

For each chessboard map, if edges and overlapping regions are more continuous and consistent, it illustrates that the registration result is more accurate. Obviously, it is observed from four CB-maps that two images are matched well by the proposed method for each data. In four CB-maps, these regions (such as river, roads, croplands, etc.) are continuous and consistent, which illustrates that two images are accurately registered. These visual results also validate that the proposed method can obtain more accurate registration results.

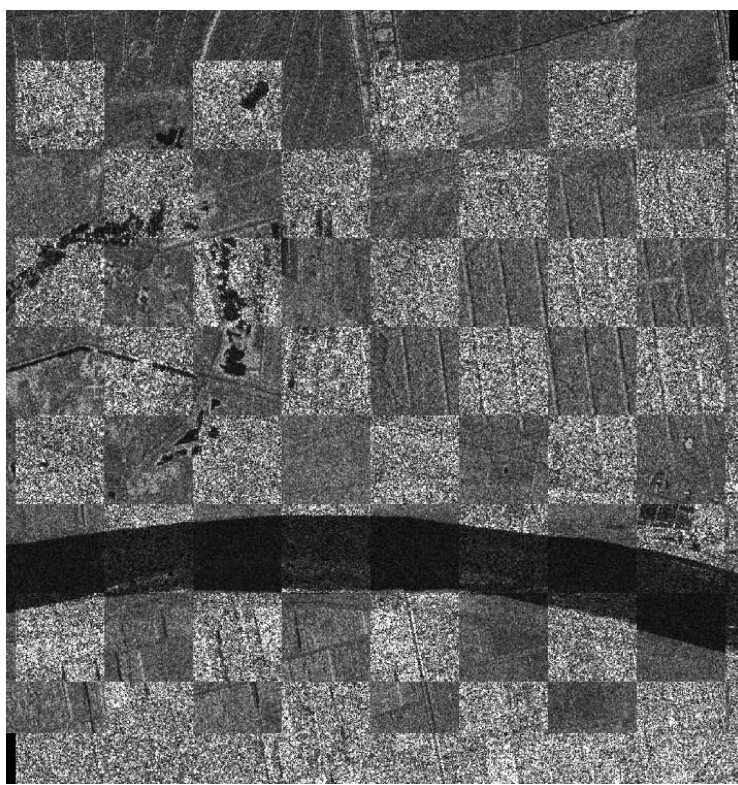

**Figure 8.** Registration CB-map for YellowR1 data with $700 \times 700$.

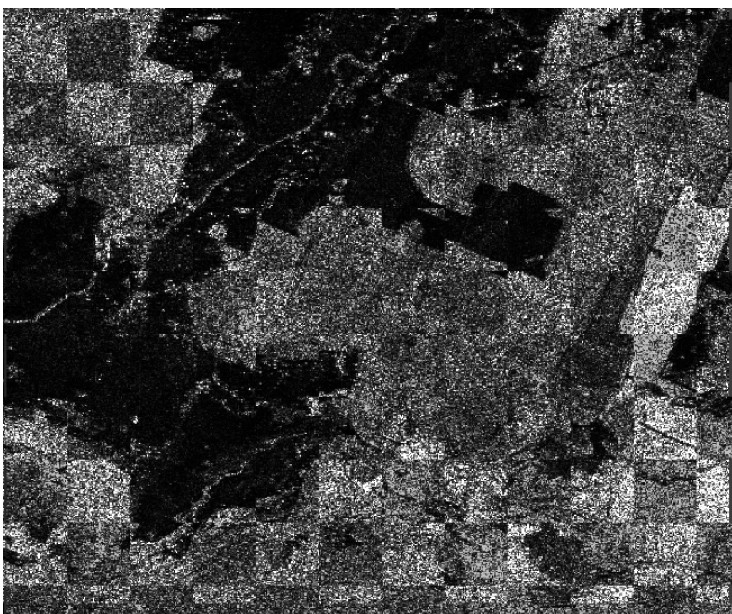

**Figure 9.** Registration CB-map for Yamba data with $350 \times 650$.

### 4.3. Analyses on the Precise-Matching Module

In this part, we give an analysis of the precise-matching module to validate its effectiveness for enhancing the registration performance. In this experiment, we show the results of two branches (the R-S sub-network and the S-R sub-network) with precise-matching and without precise-matching. $RMS_{all}$ is used as the quantitative indicator, and Table 5 shows the experimental results. From Table 5, it is seen that the accuracy without precise-matching is improved by using precise-matching for four datasets regardless of the R-S network or the S-R network. It indicates that our designed precise-matching module is effective for finding more accurate matched points.

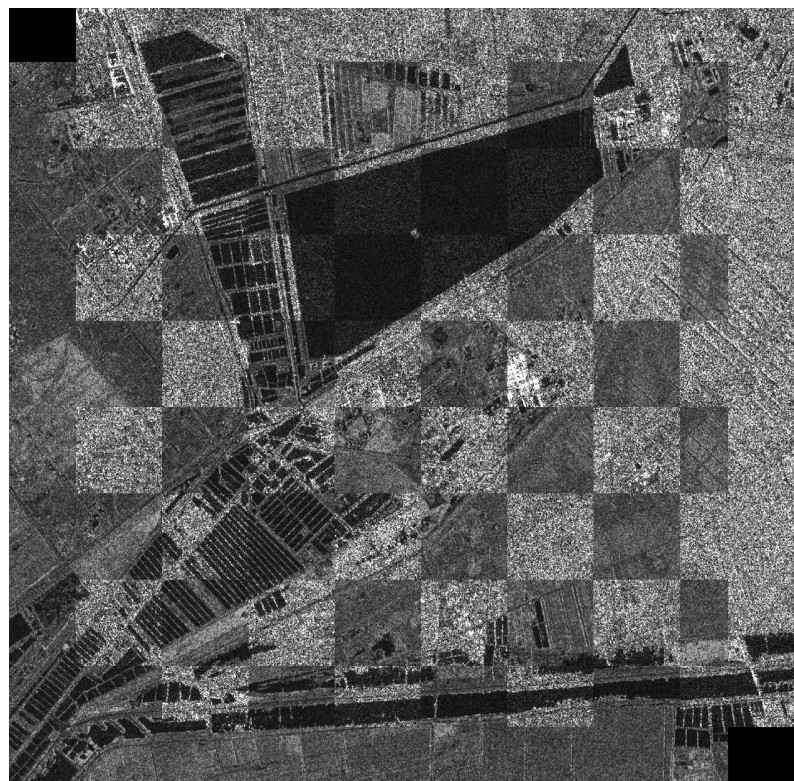

**Figure 10.** Registration CB-map for YellowR2 data with $1000 \times 1000$.

**Table 5.** A performance comparison ($RMS_{all}$) between with and without precise-matching.

| Datasets | Branch | Without Precise-Matching | With Precise-Matching |
|---|---|---|---|
| Wuhan | R→S | 0.4598 | **0.4579** |
| | S→R | 0.4620 | **0.4590** |
| YellowR1 | R→S | 0.5798 | **0.5525** |
| | S→R | 0.5585 | **0.5535** |
| YAMBA | R→S | 0.4788 | **0.4960** |
| | S→R | 0.4858 | **0.4763** |
| YellowR2 | R→S | 0.5253 | **0.5185** |
| | S→R | 0.5093 | **0.4960** |

Additionally, in order to verify the effectiveness of the precise-matching module more intuitively, we show some visual results. First, nine points were selected from the reference of Yellow River R1 data, and the obtained matched-points corresponding to nine points before and after using the precise-matching module are given. Then, their corresponding sub-images are captured from the reference image. Figure 11 shows the comparison of sub-images corresponding to nine matched-points pairs obtained by the proposed method without precised-matching module and with precise-matching module, where the locations of nine points in the reference image are given and the matched results with precised-matching module are labeled in the red box. From Figure 11, it is observed that the sub-images corresponding to the matched-points obtained by the proposed method with the precise-matching module are more similar to the original sub-images of nine points than without the precise-matching module. It also illustrates that the precise-matching is effective for improving the performance of our method.

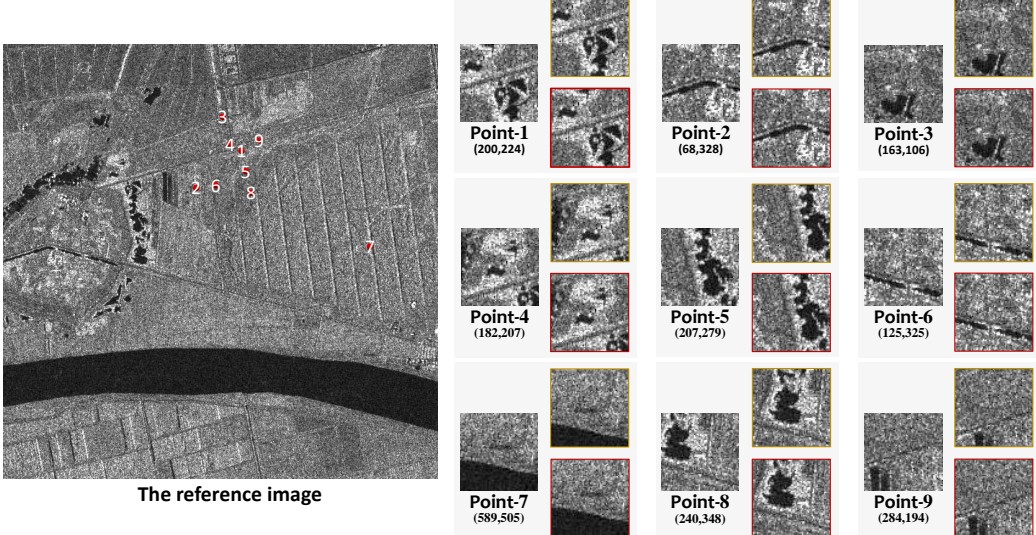

**Figure 11.** The comparison of sub-images corresponding to matched-points obtained by the proposed method without precised-matching module and with precise-matching module. For each point, the left sub-image corresponds to that point, the right top sub-image corresponds to its matched-point obtained by the proposed method without precised-matching module, and the right under sub-image (labeled in the red box) corresponds to the matched results with precised-matching module.

### 4.4. Analyses on the Double-Transformation Network

In this part, we give an analysis of the proposed double-transformation network. Figure 12 shows the comparison of the root mean square error (RMSE, $RMS_{all}$) obtained by the proposed method (with two branches) and only using one branch (the R-S network or the S-R network) for the four datasets. From Figure 12, it is seen that the proposed method obtains more registration results than only using the single network, which indicates that our double-transformation network can seek more accurate matched-points between two images compared with two single networks (the R-S network or the S-R network), since these matched-points pairs are obtained based on two multi-classification models which are trained in two feature spaces.

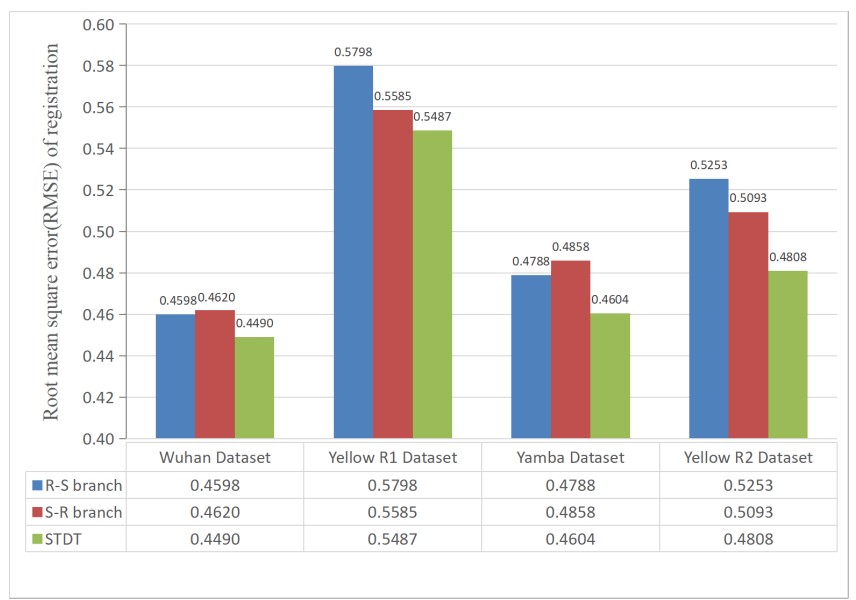

**Figure 12.** The comparison of the proposed double-transformation network to two single branches (the R-S branch and the S-R branch).

## 5. Discussion

Based on the above experimental results and analyses, it is demonstrated that the proposed method can obtain more accurate registrations of SAR images than other existing methods. According to our analyses, the reasons mainly focus on several items: First, the proposed method utilizes key points as the independent classes to handle SAR image registration, which effectively avoids the defects of traditional two-classification model for SAR image registration. Compared with existing methods, it is novel to directly use key points as independent classes to construct the multi-classification model. Additionally, key points are easily obtained from an image, which means that there are more corresponding training samples when there are more key points. Second, considering the categories of training and testing sets in SAR image registration, the proposed method is designed with a double-transformation network to construct the multi-classification model. Specially, two sub-networks effectively achieve the complementation of obtaining predicted matched-points, since two sub-networks are trained based on different key points (categories). Third, a precise-matching module is designed based on the nearest points to modify the predictions of two sub-networks and obtain more consistent matched-points. Moreover, since the detection of key points is not key for the proposed method, a simple method is used to detect the key points from two SAR images in our model, whereas it does not mean that other advanced methods are not adaptive for our model. In actuality, these methods can be used to detect the key points.

In addition, the Swin-Transformer is used as the basic network to construct the training model of the proposed method. Therefore, we also make a classification performance comparison of Swin-Transformer with three different basic networks, including VGG16 [45], ResNet50 [46], and ViT [39], which may be helpful to select which basic network to use as the basic classification model for researchers. In this part, the four networks have the same hyperparameters and the number of training iterations, and the accuracy rate (*Acc*) and the running time (*Time*) are used as the verifiable indicators to analyze the classification performances of four basic networks.

Table 6 shows the experimental results of the four networks on the Yellow River R1 data and the Wuhan data, where the best results are in bold. From Table 6, it is obvious that Swin-Transformer obtains higher accuracy rates of classification and has shorter running times compared with the other three basic networks. It indicates that the proposed method using Swin-Transformer as the basic classification network is more effective and more suitable for SAR image registration.

**Table 6.** Quantitative comparisons of the four classification networks.

| Datasets | Performance | VGG16 | ResNet50 | ViT | Swin-Transformer |
|----------|-------------|-------|----------|-----|------------------|
| YellowR1 | *Acc* (%) | 87.13 | 89.32 | 89.59 | **92.74** |
| | *Time* (m) | 47 | 38 | 42 | **31** |
| Wuhan | *Acc* (%) | 89.26 | 92.71 | 91.10 | **94.83** |
| | *Time* (m) | 19 | 13 | 28 | **10** |

## 6. Conclusions

In this paper, we propose a multi-class double-transformation network based on Swin-Transformer for SAR image registration. Different from most existing methods, the proposed method considers each key point of an SAR image as a category to construct the multi-classification model for SAR image registration. To find the matched-points between two images, we design a double-transformation network with two branches, which are trained based on multiple key points, respectively, from the reference image and the sensed image. In particular, while constructing the training and testing samples, one image is used as the base image to capture the sub-images corresponding to the key points by using an initial registration transformation, which can effectively weaken the inherent diversity between SAR images. Furthermore, a precise-matching module is designed to

search for more accurate matched-points by eliminating the inconsistent matched-points between the results obtained by two branches. Finally, experimental results show that the proposed method can achieve more accurate registration than the compared methods.

**Author Contributions:** Conceptualization, X.D. and S.M.; methodology, X.D., J.Y. and S.M.; software, J.Y. and S.L.; validation, X.D., S.G. and L.J.; formal analysis, S.M.; investigation, L.J.; resources, S.M.; data curation, Y.Z.; writing—original draft preparation, S.M., S.L. and J.Y.; writing—review and editing, X.D., S.M. and S.G.; visualization, S.L.; supervision, L.J.; project administration, X.D.; funding acquisition, S.M. All authors have read and agreed to the published version of the manuscript.

**Funding:** This research was funded by the State Key Program of National Natural Science of China (No. 62234010), the National Natural Science Foundation of China (No. 61806154 and No. 62102296), the China Postdoctoral Science Foundation Funded Project (No. 2019M653565), and Natural Science Foundation of Shaanxi Province (Grant No. 2022JQ-661).

**Data Availability Statement:** The public datasets used in experiments can be accessed at the following addresses: Bern Flood Data: https://github.com/summitgao/SAR_Change_Detection_CWNN, accessed on 1 June 2023.

**Conflicts of Interest:** The authors declare no conflict of interest.

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
