# Peer review of "Multi-Class Double-Transformation Network for SAR Image Registration"

_remotesensing, doi:10.3390/rs15112927_

Round 1

Reviewer 1 Report (Previous Reviewer 1)

After careful reviewing, I believe that the authors have addressed all my questions, and I have no other extended questions.

I believe the authors have polished their English

Author Response

Thanks a lot for your comments, and we believe your previous comments are very helpful for improving the quality of our manuscript. 

Reviewer 2 Report (Previous Reviewer 2)

The authors have satisfactorily addressed all previous concerns. I recommend publishing the paper in its current form.

Author Response

Thanks a lot for your comments, and we believe your previous comments are very helpful for improving the quality of our manuscript.

Reviewer 3 Report (New Reviewer)

The authors devised a SAR image registration method from the perspective of multi-classification. Existing methods tend to construct the positive instances (matched-point pairs) via binary-classification, whereas this work presents a double transformation network to establish a multi-classification model. Extensive experiments suggest that the proposed method can achieve a better performance for SAR image registration compared to selected conventional methods.

Adopting key point-wise independent classes to handle SAR image registration can address the characteristics of SAR imagery on essentially diverse points, which avoids the defects of conventional SAR image registration.  

As key point extraction is a key step to the proposed method, the authors are yet to explain in detail why SAR-SIFT has been selected.

It is noteworthy that the authors conducted experiments to compare the proposed method and AdaSSIR, which has a close theoretical foundation. The authors may consider comparing SSTA-Net (reference as follows), which is novel and fundamentally different from the proposed method, for a more thorough analysis.

[1] Zou B , Li H , Zhang L. Self-Supervised SAR Image Registration With SAR-Superpoint and Transformation Aggregation[J]. TGRS. 61:1-15, 2023.

Discussion on the contributions is a bit lengthy and yet to strongly emphasize the key ones. The authors may rephrase or polish this part more compactly.

Typos need addressing (e.g., the missing citation in line 322).

Author Response

Thanks for your positive and encouraging comments. According to your comments, we have added the cite of SSTA-Net. In the SSTA-Net method, it proposed an interesting method to detect rich and stable feature points, which generates the required self-training dataset based on the synthetic dataset. Actually, its method can be used in our method to obtain better key points. In our method, key points are directly considered as the independent categories to construct the multi-classification model, but detecting key points is not the key of the proposed method, and thus we only used a simple method (SIFT) to obtain key points from two SAR images. If the method of detecting key points is better, we think the proposed method may be better. Whereas, we also hope that our proposed method can obtain better performance when a simply method is used to detect key points, and it is more robustness. Hence, the detection of key points is not the key of our method. We think your suggestion is very helpful for us, which bring some new inspirations. Due to the limited modification time, the related experiments were not implemented and added in our revised manuscript. In our further works, we will add some experiments and analyses about SSTA-Net. Additionally, according to your suggestions, we have revised the description of our contributions, and the revised parts are labeled in blue in our revised manuscript. 

This manuscript is a resubmission of an earlier submission. The following is a list of the peer review reports and author responses from that submission.

Round 1

Reviewer 1 Report

In this manuscript, the authors proposed a double transformation network for SAR image registration, I have some opinions:

1. In my opinions, the contribution of this manuscript is limited. The main part of this manuscript is section 3.2 and 3.3. However, the contribution is proposed a registration strategy, tech improvements is less.

2. Some comparisons are less meaningful, take Table 6 as example, it is common for most of the researchers that swim transformer surpasses the other compared schemes in many fields, so this comparison provide less meaningful information.  

3. Line 288: the performance should be detailed instead of just reference. 

4. Some performance matric of SAR is not used in the experiment, which makes the results inconvincible.  

5. I suggest the contribution should be pointed out more clearly.

Reviewer 2 Report

The paper is overall well-written, especially the part of real data analysis and comparison with other state-of-the-art method. However, I have a few suggestion to improve the quality of the paper

Line 73-85: I suggest using bullet points to express the contributions clearly

Line 29: Omit the word "respectively"

Line 86: Replace the word "chapter" with "paper"

Line 95: Use past tense for "propose"

A general comment: It would be much more interesting to mention some benefits of "SAR image registration" in complex SAR-based scenarios. For example, Open set recognition. The relationship between SAR Image Registration and Open Set Recognition lies in the potential use of SAR Image Registration for improving the performance of Open Set Recognition. In some cases, SAR Image Registration can be used to preprocess the input SAR images, aligning them to a common reference frame and correcting any geometric distortions. This can improve the performance of Open Set Recognition algorithms by providing more consistent and accurate input data, which can lead to better feature representations and improved classification results. To include a brief description on open set recognition, the following paragraph from a paper of the same journal (Remote Sensing, 2022) can be mentioned: Open set recognition, which refers to the ability to detect unknown targets and contributes to reducing the error rate and enhancing the precision in classification tasks, may be a desirable feature, especially in military applications when false alarms need to be kept under control [X]: Proportional Similarity-Based Openmax Classifier for Open Set Recognition in SAR Images

Line 288: It is better to mention the names of the performance indicators here in the text, rather than using the reference. The mathematical formulations can be found in the reference for interested readers, but the names are vital to be mentioned here. For example, Dice Similarity Coefficient (DSC), which is a binary similarity measure used to evaluate the performance of SAR image registration algorithms in the presence of binary or semi-binary features, is not mentioned among the indicators?

Line 359: RMSE

Figure 11:provide y-label

Reviewer 3 Report

1.       Abstract does not representing the proposed methodology. Add precise contributions clearly in the abstract along with the findings.

2.       Many existing equations are presented without suitable references such as equations 1 to 3.

3.       How authors have reduced the educing the computational complexity in the W-MSA module. Derive the equation 4.

4.       Visual analyses should be better presented and discussed.

5.       How authors have selected the hyper-parameters of the proposed model? If trail and error based then authors should mention in future scope do consider “Evolving fusion-based visibility restoration model for hazy remote sensing images using dynamic differential evolution”

6.       Add more approaches for quantitative analyses.

7.       How authors have obtained the M* t training set?

8.       Provide more mathematical details regarding the Swin-Transformer